# Mice Immunized with the Vaccine Candidate HexaPro Spike Produce Neutralizing Antibodies against SARS-CoV-2

**DOI:** 10.3390/vaccines9050498

**Published:** 2021-05-12

**Authors:** Chotiwat Seephetdee, Nattawut Buasri, Kanit Bhukhai, Kitima Srisanga, Suwimon Manopwisedjaroen, Sarat Lertjintanakit, Nut Phueakphud, Chatbenja Pakiranay, Niwat Kangwanrangsan, Sirawat Srichatrapimuk, Suppachok Kirdlarp, Somnuek Sungkanuparph, Somchai Chutipongtanate, Arunee Thitithanyanont, Suradej Hongeng, Patompon Wongtrakoongate

**Affiliations:** 1Department of Biochemistry, Faculty of Science, Mahidol University, Bangkok 10400, Thailand; chotiwat.see@student.mahidol.ac.th (C.S.); nattawut.bua@alumni.mahidol.ac.th (N.B.); kitima.sri@mahidol.ac.th (K.S.); sarat.lej@student.mahidol.ac.th (S.L.); nut.phe@alumni.mahidol.ac.th (N.P.); chatbenja.pai@student.mahidol.ac.th (C.P.); 2Department of Physiology, Faculty of Science, Mahidol University, Bangkok 10400, Thailand; kanit.bhu@mahidol.ac.th; 3Department of Microbiology, Faculty of Science, Mahidol University, Bangkok 10400, Thailand; swiboonut@gmail.com (S.M.); arunee.thi@mahidol.edu (A.T.); 4Department of Pathobiology, Faculty of Science, Mahidol University, Bangkok 10400, Thailand; niwat.kan@mahidol.ac.th; 5Chakri Naruebodindra Medical Institute, Faculty of Medicine Ramathibodi Hospital, Mahidol University, Samut Prakan 10540, Thailand; sirawat.sri@mahidol.ac.th (S.S.); suppachok.kir@mahidol.ac.th (S.K.); somnuek.sun@mahidol.ac.th (S.S.); schuti.rama@gmail.com (S.C.); 6Department of Pediatrics, Faculty of Medicine Ramathibodi Hospital, Mahidol University, Bangkok 10400, Thailand; suradej.hon@mahidol.ac.th; 7Department of Clinical Epidemiology and Biostatistics, Faculty of Medicine Ramathibodi Hospital, Mahidol University, Bangkok 10400, Thailand; 8Center for Neuroscience, Faculty of Science, Mahidol University, Bangkok 10400, Thailand

**Keywords:** HexaPro, spike, SARS-CoV-2, vaccine

## Abstract

Updated and revised versions of COVID-19 vaccines are vital due to genetic variations of the SARS-CoV-2 spike antigen. Furthermore, vaccines that are safe, cost-effective, and logistic-friendly are critically needed for global equity, especially for middle- to low-income countries. Recombinant protein-based subunit vaccines against SARS-CoV-2 have been reported using the receptor-binding domain (RBD) and the prefusion spike trimers (S-2P). Recently, a new version of prefusion spike trimers, named HexaPro, has been shown to possess two RBD in the “up” conformation, due to its physical property, as opposed to just one exposed RBD found in S-2P. Importantly, this HexaPro spike antigen is more stable than S-2P, raising its feasibility for global logistics and supply chain. Here, we report that the spike protein HexaPro offers a promising candidate for the SARS-CoV-2 vaccine. Mice immunized by the recombinant HexaPro adjuvanted with aluminum hydroxide using a prime-boost regimen produced high-titer neutralizing antibodies for up to 56 days after initial immunization against live SARS-CoV-2 infection. Also, the level of neutralization activity is comparable to that of convalescence sera. Our results indicate that the HexaPro subunit vaccine confers neutralization activity in sera collected from mice receiving the prime-boost regimen.

## 1. Introduction

The coronavirus disease 2019 (COVID-19), caused by the novel coronavirus, severe acute respiratory syndrome coronavirus 2 (SARS-CoV-2), is a current global pandemic. The incidence for this pandemic reported by World Health Organization (WHO) on 31 March 2021, has included 130 million cumulative confirmed cases and over 2.84 million deaths globally. There is an urgent need for preventative vaccines and therapeutics. SARS-CoV-2 is an enveloped, single-stranded RNA virus. Its genome encodes four structural proteins comprising of the spike (S), membrane glycoprotein (M), envelope (E), and nucleocapsid (N) proteins. The spike protein mediates viral entry by binding to the host receptor angiotensin-converting enzyme 2 (ACE2) via the receptor-binding domain (RBD). This interaction triggers a substantial conformational alteration of the spike from a prefusion conformation to a highly stable postfusion conformation [1,2,3,4]. The spike protein can induce the production of neutralizing antibodies in patients, indicating its immunogenic property. Thus, it has been widely adopted for vaccine development. However, the ongoing COVID-19 pandemic has led to SARS-CoV-2 spike variants with serious concerns such as D614G, N501Y, E484K, and 69/70 deletion [5,6,7]. Some of these variants can be highly transmissible and capable of escape vaccine-induced neutralizing antibody responses [8].

A key strategy for vaccine design against coronaviruses SARS-CoV and MERS-CoV has aimed at stabilizing the metastable prefusion conformation of the spike protein homologs [9,10]. The prefusion stabilization has been achieved with two consecutive proline substitutions (S-2P) in a turn between the central helix and heptad repeat 1 (HR1). These S-2P variants, together with a C-terminus foldon trimerization domain, have been shown as a superior immunogen [10]. As a consequence, the SARS-CoV-2 S-2P has been employed in currently used vaccines including mRNA-1273 [11], BNT162b2 [12], and ChAdOx1 [13].

In this work, we aim to provide a proof-of-concept of a recently published prefusion-stabilized spike ectodomain, namely HexaPro, developed by McLellan and colleagues [14] as a potential COVID-19 subunit vaccine. We show that the HexaPro subunit vaccine administered with aluminum hydroxide adjuvant in mice elicits a strong neutralizing antibody response against SARS-CoV-2. This finding holds a promise towards a next-generation coronavirus vaccine development using the HexaPro spike protein.

## 2. Materials and Methods

### 2.1. Ethics Statement

Mouse experiments were performed under the Animal Ethics approved by Faculty of Science, Mahidol University (MUSC63-016-524). PCR-confirmed COVID-19 patients (*n* = 58) were hospitalized at Chakri Naruebodindra Medical Institute, Faculty of Medicine Ramathibodi Hospital, Mahidol University. Serum specimens were collected from patients 14–30 days post-infection. The study protocol and human ethics were approved by Faculty of Medicine Ramathibodi Hospital (COA. MURA2020/568).

### 2.2. Expression and Purification of HexaPro Subunit Vaccine

The mammalian expression plasmid containing SARS-CoV-2 HexaPro spike was obtained from Addgene (Addgene plasmid # 154754; http://n2t.net/addgene:154754; RRID: Addgene_154754; accessed on 1 December 2020). HEK293T cells were transiently transfected with the HexaPro plasmid by calcium phosphate transfection. Cells and culture medium were separated by centrifugation. The supernatant was concentrated with Amicon^®^ Ultra–15 Ultrace–30K centrifugal filter unit (MERCK). Cell protein contents were extracted with a lysis buffer composed of 50 mM sodium phosphate, 300 mM NaCl, 20 mM imidazole, 1X CompleteTM EDTA-free protease inhibitor cocktail (Roche), 1 mM Phenylmethylsulfonyl fluoride (PMSF, Sigma-Aldrich, St. Louis, MO, USA), and 1% Triton-X (Sigma-Aldrich). Protein extracts were filtered through a 0.22 µm NalgeneTM syringe filter (Thermo ScientificTM). S HexaPro protein was then purified with HisTrap HP (cytiva) equilibrated with a buffer composed of 50 mM Sodium phosphate, 300 mM NaCl, and 20 mM imidazole. Fractions containing HexaPro were pooled and exchanged to phosphate-buffered saline (PBS). Purified protein was digested with HRV3C protease to remove purification tags. The protein was further purified with Sephacryl S-300 HR (GE Healthcare, Chicago, IL, USA) with PBS. Fractions that contained HexaPro protein were pooled and analyzed with SDS-PAGE and Western blot against the SARS-CoV-2 RBD protein (Sino Biological, Cat#40592-T62) or pooled convalescent sera. The purified protein was kept at −80 °C until use.

### 2.3. Immunofluorescence Staining

HeLa cells were transiently transfected with the plasmid encoding HexaPro using lipofectamine 3000 (Invitrogen, Cat#L3000008, Carlsbad, CA, USA). Cells were fixed with 4% PFA and were incubated with either a polyclonal antibody against the SARS-CoV-2 RBD protein (Sino Biological, Cat#40592-T62) or a monoclonal antibody against the SARS-CoV-2 S1 protein (MyBioSource, Cat#MBS434277, San Diego, CA, USA). A goat anti-rabbit secondary antibody (IgG) conjugated with Alexa Fluor 594 (Invitrogen, Cat#A-11037) or a goat anti-mouse secondary antibody (IgG) conjugated with Alexa Fluor 488 (Invitrogen, Cat#A-11029) was used for visualization under a fluorescence microscope. For convalescent serum staining, cells were incubated with heat-inactivated serum and visualized with a goat anti-human secondary antibody (IgG) conjugated with FITC (Abcam, Cat#ab97224, Cambridge, UK).

### 2.4. Mouse Immunization

Female C57BL/6 mice (7–9 weeks old, *n* = 3 per group) were ordered from Nomura Siam International. Mice were given a prime-boost immunization intramuscularly (IM), spaced three weeks apart. For antigen formulation, SARS-CoV-2 S HexaPro protein (1 µg for the first dose and 5 µg for the booster dose) was mixed with 100 µg of aluminum hydroxide (Invivogen, Cat#vac-alu-250). Serum was collected for analysis on study days 14, 35, and 56 after the initial immunization.

### 2.5. Microneutralization Assay

Heat-inactivated sera at 56 °C for 30 min were two-fold serially diluted, starting with a dilution of 1:10. The serum dilutions were mixed with equal volumes of 100 TCID50 of SARS-CoV-2. After 1 h of incubation at 37 °C, 100 μL of the virus–serum mixture at each dilution was added in duplicate to Vero E6 cell monolayers in a 96-well microtiter plate. The last two columns are set as virus control, cell control, and virus back-titration. The plates were incubated at 37 °C in 5% CO_2_ in a humidified incubator. After two days of incubation, the medium was discarded, and the cell monolayer was fixed with cold fixative (1:1 methanol:acetone) for 20 min on ice. Viral protein in the virus-infected cells was detected by ELISA assay. The cells were washed three times with PBST before blocking with 2% BSA for 1 h at room temperature. After washing, the viral nucleocapsid was detected using 1:5000 of SARS-CoV/SARS-CoV-2 Nucleocapsid monoclonal antibody (Sino Biological, Cat#40143-R001) by incubation at 37 °C for 1 h. After removing the detection antibody, 1:2000 HRP-conjugated goat anti-rabbit polyclonal antibody (Dako, Denmark A/S, Cat#P0448) was added, and the plate was incubated at 37 °C for 1 h. After washing, the TMB substrate (KPL, Cat#5120-0075) was added. After 10 min incubation, the reaction was stopped by the addition of 1N HCl. Optical density (O.D.) at 450 and 620 nm was measured by a microplate reader (Tecan Sunrise).

The virus neutralization endpoint titer of each serum was calculated using the following equation:*X = [(average A450 of virus control wells) − (average A450 of cell control wells)]/2 + (average A450 of cell control wells)*(1)

The reciprocal of the highest dilution of serum with O.D. values less than *X* is considered positive for neutralization activity. Serum samples that tested negative at a dilution of 1:10 were assigned an NT titer of <10. The serum that tests positive at 1:10 dilution will be reported as the NT titer of 20.

Each sample was carried out in duplicate. Live SARS-CoV-2 viruses at passage 3 or 4 and Vero E6 cells at the maximum passages of 20 were employed. The activities with live viruses were carried out in a certified biosafety level 3 facility.

## 3. Results

### 3.1. Expression and Purification of Recombinant SARS-CoV-2 HexaPro Spike Protein

The prefusion-stabilized HexaPro construct (Figure 1A) encoding the spike ectodomain of SARS-CoV-2 with proline substitution at residues 817,892,899, 942,986, and 987, “GSAS” substitution at residues 682–685 (the furin cleavage site), and C-terminal foldon trimerization motif [14] was used to produce the HexaPro subunit vaccine in HEK293 cells. Transient transfection of HexaPro-encoding plasmid into the cells resulted in recombinant protein expression in the culture supernatant. The recombinant HexaPro protein was purified by Ni-NTA chromatography, followed by size-exclusion chromatography. The purity of the purified recombinant HexaPro was ascertained by SDS-PAGE (Figure 1B). Using pooled convalescence sera from COVID-19 patients and Western blot analysis, we show that the HexaPro spike is immunogenic.

To confirm the immunogenicity of the HexaPro spike, immunofluorescence staining was then performed and this validated that antibodies could detect the spike structural variant against SARS-CoV-2 spike protein, which was also validated by the pooled convalescent sera (Figure 2). These results illustrate the potential of the HexaPro recombinant protein as a subunit vaccine.

### 3.2. Neutralization of SARS-CoV-2 by Sera Collected from HexaPro-Immunized Mice

The HexaPro subunit vaccine was then evaluated for its immunization activity via neutralization of SARS-CoV-2 by immunized mouse sera. An immunization protocol of low priming dose followed by high booster dose was followed. On day 0, C57BL/6 mice were prime-immunized with 1 µg of HexaPro adjuvanted with aluminum hydroxide (100 µg) via intramuscular administration. On day 21, the mice were boost-immunized with 5 µg of HexaPro (Figure 3A). The microneutralization assay using live SARS-CoV-2 infection in Vero E6 cells was performed with sera collected on days 14, 35, and 56 after initial immunization. At 14 days after the priming dose, we did not observe a significant neutralizing activity in vaccinated mice (Figure 3B). However, sera from vaccinated mice collected 14 days after the booster dose elicit high neutralization titers. Furthermore, the level of neutralization activity was sustained at least 56 days after the initial immunization (Figure 3B). Together, our results indicate that the HexaPro subunit vaccine adjuvanted with aluminum hydroxide confers neutralization activity in sera collected from mice receiving the prime-boost regimen.

## 4. Discussion

In this report, we observed a potent SARS-CoV-2 neutralizing activity delivered by the subunit vaccine HexaPro spike; four amino acids of which were substituted by McLellan and colleagues into beneficial prolines leading to a more stable spike variant [14]. Specifically, the amino acid substitution was engineered within the S2 domain of the original S-2P spike [2]. This novel prefusion variant possesses 30% of the spike trimers being an “up” conformation with two exposed RBD, as opposed to just one exposed RBD found in S-2P. Due to its enhanced stability, the HexaPro spike has been proposed for its potential as a COVID-19 vaccine.

Using alum in vaccination has been shown to enhance activation of inflammatory dendritic cells and T-cell responses [15,16,17]. In a phase 1 trial, alum was employed with the inactivated SARS-CoV-2 vaccine BBV152 [18] as well as in ongoing clinical trials of COVID-19 vaccines, including subunit vaccines (NCT04522089; NCT04527575; NCT04683484; NCT04742738) and an inactivated SARS-CoV-2 vaccine (NCT0464148). In addition, we also adopted a regimen of prime-boost immunization using a low priming dose followed by a high booster dose. A number of studies and clinical trials have demonstrated that a lower priming dose, followed by a subsequent higher booster dose, can induce more significant levels of the immune response [19,20,21], including the COVID-19 vaccine ChAdOx1 [13]. Importantly, effector cells are adversely induced by higher doses of antigen at prime immunization. On the other hand, immune memory cells are promisingly induced by lower doses at prime immunization, making this regimen suitable for long-term immunological memory [22].

Prefusion-stabilized spike proteins have been reported to facilitate their ectopic expression possibly by avoiding transition into a postfusion structure. Specifically, the HexaPro is more stable than its S-2P counterpart, which is difficult to be ectopically expressed and produced in mammalian cells [14]. Moreover, using biophysical assays, HexaPro has been shown to be more resistant to cold-induced denaturation than the previous generation S-2P variant [23]. Altogether, due to its highly stable conformation and feasible production, this HexaPro spike is potentially logistically applicable, and should be further developed into a COVID-19 vaccine and exploited for its efficacy in viral challenge studies using different SARS-CoV-2 genetic variants. Moreover, this HexaPro variant is suitable for development of next-generation mRNA, DNA, and viral vector vaccines. Indeed, an ongoing phase I/II clinical trial has already been initiated by the Government Pharmaceutical Organization of Thailand and Mahidol University for a viral vector vaccine, NDV-HXP-S, which utilizes the HexaPro spike as the SARS-CoV-2 antigen (NCT04764422).

## 5. Conclusions

We provide a proof-of-concept which indicates that the HexaPro subunit vaccine confers neutralization activity in sera collected from mice receiving the prime-boost regimen. We support the use of this HexaPro spike variant for next-generation COVID-19 vaccines.

## Figures and Tables

**Figure 1 vaccines-09-00498-f001:**
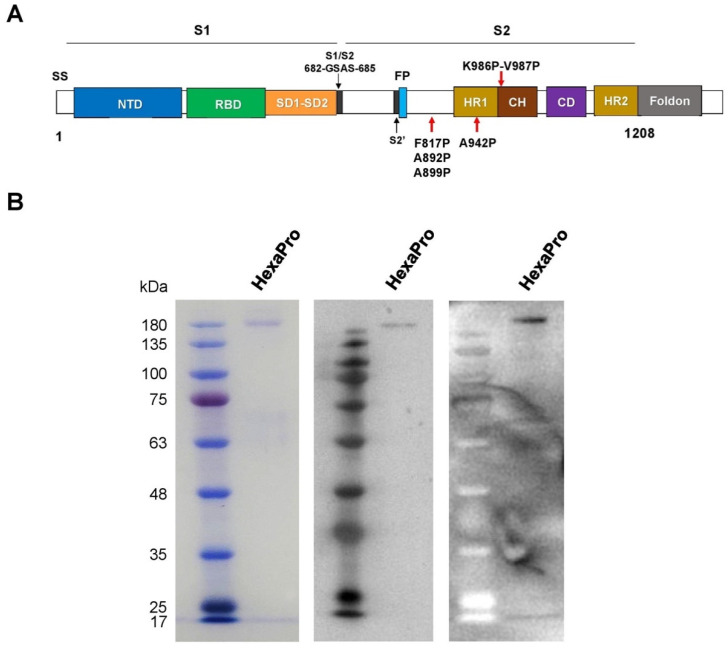
The recombinant SARS-CoV-2 HexaPro spike protein. (**A**) Schematic representation of the prefusion-stabilized SARS-CoV-2 HexaPro ectodomain showing the S1 and S2 subunits. Four additional proline substitutions from the S-2P construct are indicated by the red arrows shown below the construct. (**B**) The HexaPro protein expressed in HEK293T cells was purified and characterized by SDS-PAGE (**left**), Western blot using a commercial anti-RBD (**middle**), and Western blot using pooled convalescence sera (**right**).

**Figure 2 vaccines-09-00498-f002:**
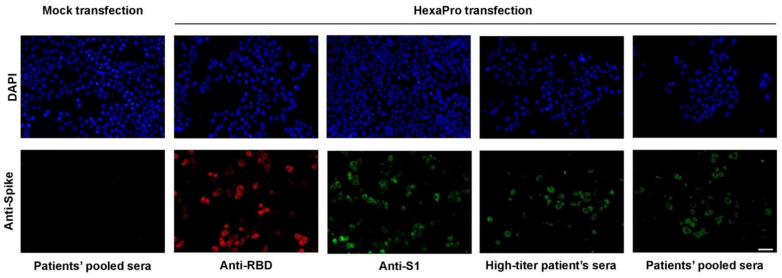
Immunofluorescence staining of the HexaPro spike expressed in HeLa cells using antibodies against spike RBD and S1 subunits, and convalescent sera derived from a patient with high-titer neutralization activity and from pooled sera. Scale bar: 50 µm.

**Figure 3 vaccines-09-00498-f003:**
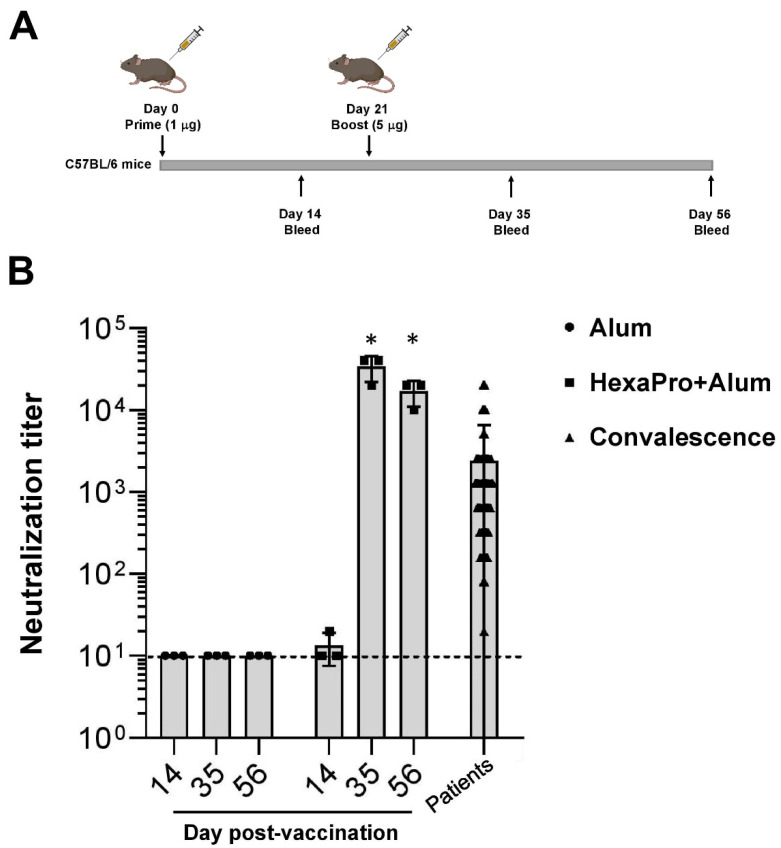
The prime-boost regimen using the recombinant HexaPro adjuvanted with aluminum hydroxide results in sera possessing neutralization activity. (**A**) C57BL/6 mice were vaccinated intramuscularly with Alum or HexaPro (1 µg) + Alum. On day 21 they then received a booster dose with HexaPro (5 µg) + Alum. (**B**) The virus neutralization endpoint titer of sera collected from mice and from convalescence sera. The dashed line shows the limit of detection. Neutralization activity on days 35 and 56 was compared to day 14. The error bars indicate the ±SD. Comparisons were performed using Student’s *t*-test (unpaired, two tail); * *p* < 0.01 (made in ©BioRender-biorender.com (accessed on 2 March 2021)).

## Data Availability

The data presented in this study is contained within the article.

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
