# Peer review of "Mice Immunized with the Vaccine Candidate HexaPro Spike Produce Neutralizing Antibodies against SARS-CoV-2"

_vaccines, 2021, doi:10.3390/vaccines9050498_

Round 1
Reviewer 1 Report
The manuscript by Seephetdee et al evaluates the antibody response of Hexfold SARS-CoV-2 vaccine with alum adjuvant in mice. The manuscript lacks rationale and does not contribute anything new or useful to the field. The study seems incomplete.The antibody response without comparing with non adjuvant vaccine doesn’t convey anything. Even if it was compared, it would not be much useful without any in-depth study. The authors should have rationale and test it appropriately. The current form of manuscript is not recommended to the editor.
Reviewer 2 Report
It is known that the S protein ectodomain from related MERS-CoV is less stable and more difficult to produce than other S proteins, and soluble constructs of the RBD have been the main focus of structural studies, antibody isolation efforts, and subunit vaccine development. This research provides the first animal immunisation results of a promising vaccine candidate proposed by McLellan et al in 2020, and shows that using a prime-boost regimen. Despite the small sample size (only n=3 mice per vaccine group) the immune sera showed high neutralizing titres that were comparable to the convalescence sera.
Questions:
In Section 2.1 (Ethics Statement) only animal ethics approval was mentioned. How about human ethics approval for the convalescence sera?
Line 183: what is “minute” neutralization activity?
Line 221: The author claimed that the HexaPro Spike demonstrates advantages such as feasible production and logistic applicability compared to other subunit vaccine constructs. However, no information or citation about the yield of the recombinant protein production or stability profile at different storage conditions (which can be easily cited from McLellan’s original paper) was given, and it was noted that the proteins produced for this research was “kept in -80°C until use”.
Line 224: Need further elaboration for the statement “mRNA, DNA and viral vector vaccines should benefit by using this HexaPro variant”.
Minor corrections:
Line 101: the brackets flanking “Sino biological” was doubled.
Line 102 and Lines 145-147: past tense should be used for these sentences.
Reviewer 3 Report
In this manuscript Seephetdee et al, use a previously published stable spike protein to induce neutralizing antibodies in mice. They establish that HexaPro can be recognized by Sars-Cov2 antibodies and that, along with aluminum hydroxide, can induce a prolonged immune response and neutralizing antibody response.
The paper is well written and logically flows. The experiments are well controlled and the conclusions are well supported.
Comments:
Using patient sera is a good control, however, more details should be included on these samples. Specifically, more details should be meantioned in the section "Neutralization of SARS-CoV-2 by Sera Collected from HexaPro-Immunized Mice". Include the detail that these are patients that these were infected patients. How much sera was used and is this comparable to the mouse samples? Also, are these patients 14-30 days post infection, post positive test, or admission to hospital?
There is no control for transfection efficiency in Figure 2 (perhaps and IRES-GFP). Though not necessary, it would be nice to compare how well antibodies recognize the antigen.
Reviewer 4 Report
The current study is a proof-of-concept of the previously reported prospective vaccine candidate HexaPro. Authors have shown that HexaPro Spike adjuvanted with aluminium hydroxide produces neutralizing antibodies against SARS-CoV-2.
The article is well structured into section and subsections and professionally written. It is within the scope of journal and will be of interest to the readers.
There are some minor suggestions to improve the article:
- Page 4, figure 1: The legend for panel (A) can be elaborated for the better understanding of the readers. The regions of S1 and S2 subunits can be mentioned in the legend. For example, SS, S’, FP, CH etc denotes ….
- Page 5, figure 2: A merged image of nucleus staining and location of protein of interest can be shown in a third row below each image. This will make it easier to identify colocalization.
- Page 8-9: A consistent referencing pattern should be used. Authors need to update some of the references. The details like volume, issue, page numbers are missing. For example, check reference - 3, 4, 10, 19, and 20.
Reviewer 5 Report
Overall, the work is well-done, the manuscript clearly written and this is an important study.
Comments:
On page #3, authors need to provide detail protocols of Mouse Immunization. 1 mg of SARS-CoV-2 S HexaPro protein for the first dose and 5 mg for the booster dose are quite large amounts. Authors also need to explain about 1st and 2nd dose amount.
Round 2
Reviewer 1 Report
The authors highlight the neutralizing antibody generated from the HexaPro protein-Alum in mice as proof-of-concept (PoC). Although, this is new information, it is not surprising and can not be deemed as PoC. Spike antigen can induce neutralizing antibody that has been demonstrated by many vaccines and vaccine candidates, there is not major difference between the protein sequence. HexaPro is stabilized and should still behave similar to existing vaccines. In the HexaPro (Science 2020) study, the protein binds to the patient sera. The present study does not provide any significant new information other than obvious. The advancement of this candidate in clinical trials in different countries is an indication that it has already qualified this PoC stage. The authors should perform some in-depth study and incorporate new information so that is useful to the scientific community.